# Quantitative Analysis of Factors Regulating Angiogenesis for Stem Cell Therapy

**DOI:** 10.3390/biology10111212

**Published:** 2021-11-20

**Authors:** Takahiro Shimazaki, Nobuhiro Noro, Kazuhiro Hagikura, Taro Matsumoto, Chikako Yoshida-Noro

**Affiliations:** 1Department of Applied Molecular Chemistry, Graduate School of Industrial Technology, Nihon University, Narashino, Chiba 275-8575, Japan; takahiro.shimazaki.m2eagle@gmail.com; 2GlaxoSmithKline Pharmaceuticals Ltd., Minato-ku, Tokyo 107-0052, Japan; nobuhiro.noro@bonac.co.jp; 3Division of Cell Regeneration and Transplantation, Department of Functional Morphology, Nihon University School of Medicine, Itabashi, Tokyo 173-8610, Japan; kazuhiro.hagikura@gmail.com (K.H.); matsumoto.taro@nihon-u.ac.jp (T.M.)

**Keywords:** HUVECs, angiogenesis, Ang-1, VEGF, flavonoid, polyphenol, DFAT

## Abstract

**Simple Summary:**

The control of angiogenesis is essential in disease treatment or regenerative medicine. We conducted a clinical study of dedifferentiated fat (DFAT) cells, a kind of mesenchymal stem cells, by applying cell transplantation therapy to induce angiogenesis in patients with severe ischemic disease. This study aimed to analyze the effect of molecules that regulate angiogenesis in vitro and clarify their molecular mechanisms for therapeutic purposes. Normal human umbilical venous endothelial cells (HUVECs) were cultured in the presence of vascular endothelial growth factor (VEGF). Recombinant human angiopoietin-1-producing cells, conditioned media, mouse DFAT cells, and antioxidant polyphenols were added to this system at various concentrations. After 11 days, the cultures were immunostained with CD31 (PECAM-1), and microscopic images were subjected to analysis (area, length, joint, and path) by using software to quantitatively analyze blood vessel formation. The expression of angiogenic markers and COX pathway genes were analyzed by RT-PCR. As a result, the dose-dependent angiogenesis-promoting effect of rAng-1-producing cells, conditioned medium, or commercially available recombinant Ang-1 were observed. DFAT cells also promoted angiogenesis, whereas polyphenols inhibited angiogenesis in a dose-dependent manner.

**Abstract:**

(1) Background: The control of angiogenesis is essential in disease treatment. We investigated angiogenesis-promoting or -suppressing factors and their molecular mechanisms. (2) Methods: Angiogenesis from HUVECs was quantitatively analyzed using the Angiogenesis Analysis Kit (Kurabo, Osaka, Japan). Human rAng-1-producing 107-35 CHO cells or mouse DFAT-D1 cells were co-cultured with HUVEC. Antioxidant polyphenols were added to the culture. Gene expression was analyzed by RT-PCR. (3) Results: The addition of rAng-1-producing cells, their culture supernatant, or commercially available rAng-1 showed a promoting effect on angiogenesis. The co-culture of DFAT-D1 cells promoted angiogenesis. Polyphenols showed a dose-dependent inhibitory effect on angiogenesis. Luteolin and quercetin showed remarkable anti-angiogenic effects. The expression of vWF, Flk1, and PECAM-1 was increased by adding rAng-1-producing cell culture supernatant. Polyphenols suppressed these genes. Apigenin and luteolin markedly suppressed α-SMA and Flk1. Resveratrol and quercetin enhanced the expression of PPARγ, and luteolin suppressed the expression of COX-1. The expression of endothelial nitric oxide synthase (eNOS), an oxidative stress-related gene, was slightly increased by luteolin. These results suggest that polyphenols induce ROS reduction. (4) Conclusions: We showed the promoting effect of Ang-1 or DFAT and the suppressing effect of polyphenols on angiogenesis and studied their molecular mechanisms. These results help control angiogenesis in regenerative therapy.

## 1. Introduction

### 1.1. Angiogenesis

Vasculogenesis and angiogenesis are the fundamental processes by which new blood vessels are formed. Vasculogenesis is the differentiation of precursor cells into endothelial cells to form a primitive vascular network [1] and mainly occurs during development. The vascular system is derived from the mesoderm and is an organ that begins forming from the earliest stage of embryogenesis when mesoderm stem cells, endothelial progenitor cells (EPCs), are thought to differentiate into vascular endothelial cells [2]. Even in adults, vascular remodeling, such as hierarchical changes and the retraction of blood vessels, occurs according to the demand for oxygen and nutrients.

Angiogenesis refers to the growth of new capillaries from preexisting blood vessels via either sprouting or non-sprouting [3]. Sprouting angiogenesis occurs when endothelial cells produce proteases to digest the extracellular matrix around blood vessels to form protrusions that migrate and proliferate to grow buds. Then, blood flow starts by the fusion of protrusions with protrusions from other blood vessels. At the same time, a new basement membrane of blood vessels is formed; endothelial cells are re-differentiated; and finally, pericytes surround the endothelial cells to create mature blood vessels. Sprouting is the characteristic mode when angiogenesis invades tissues without vascular distribution.

On the other hand, non-sprouting angiogenesis is seen in the tissues where blood vessels are already distributed. It is a mechanism by which the lumen expands due to the proliferation of endothelial cells (a partition wall is formed in the lumen) and blood vessels separate into two or more blood vessels. This phenomenon occurs when the blood vessel density in tissues increases. These new blood vessels are composed of only endothelial cells, but eventually, vascular smooth muscle cells and the extracellular matrix gather in capillaries and surround the endothelial cells to form the mature walls of large blood vessels such as arteries (Figure 1) [4].

### 1.2. Disease and Angiogenesis

Angiogenesis is a concept established by Folkman in 1971 [5]. He found that cancer development depends on blood vessels invading from the surroundings. The factors released from cancer or wound tissues in hypoxia induce angiogenesis. Hanahan and Weinberg proposed six features of cancer [6]: the acquisition of autonomous growth signals, the avoidance of growth suppression signals, the release from apoptotic cell death, unlimited proliferative capacity, angiogenesis, infiltration, and metastasis. Tumor blood vessels supply oxygen and nutrients and support metastasis in distant organs [6]. Cancer is known to secrete vascular growth factors and inducers. Angiogenic factors bind to receptors on the surface of vascular endothelial cells to initiate cell stimulation, and proteolytic enzymes are produced from the stimulated vascular endothelial cells to degrade the basement membrane, resulting in new blood vessels from existing ones [7]. Then, vascular endothelial cells migrate, proliferate, form lumens, and stabilize [4]. New blood vessels are formed through such a process. In recent years, molecular biological analysis revealed many molecules involved in each process and their inhibitors. Using these factors, molecular-targeted drugs for cancer that suppress angiogenesis are also being developed [8].

Other known diseases associated with angiogenesis are proliferative diabetic retinopathy, age-related macular degeneration, and retinopathy of prematurity.

### 1.3. Regenerative Medicine and Angiogenesis

In regenerative medicine, when transplanting artificially constructed three-dimensional (3D) tissue, a poor vascular network limits the supply of oxygen and nutrients to tissue cells, resulting in cell death and a low tissue survival rate. Therefore, it is necessary to develop a system for constructing a vascular network in 3D tissue. Hypoxia-responsive gene expression systems have been developed to produce the angiogenic vascular endothelial growth factor (VEGF) and improve hypoxia and malnutrition in artificial 3D tissues [9,10].

On the other hand, stem cell transplantation therapy is performed for ischemic diseases associated with the stenosis of peripheral arteries such as arteriosclerosis obliterans (ASO) and thromboangiitis obliterans (TAO). The mainstream treatment strategy for these diseases has been medical treatment methods using drug therapy or surgical treatment methods such as bypass surgery and intervention treatment. However, in the case of severe ischemic disorders, it cannot prevent lower limb amputation. We are conducting a clinical study of cell transplantation therapy using Dedifferentiated adipocytes: DFAT cells in patients with severe ischemic disease [11]. DFAT cells established by culture from the patient’s fat tissue are locally transplanted into the lower limbs. Although it has been confirmed by animal experiments that transplanted DFAT cells also participate in some vascular differentiation, it is suggested that the main factor is that DFAT cells produce various cytokines and promote the self-regeneration of blood vessels [12].

### 1.4. Factors Involved in Angioplasty

With development and differentiation, several factors continuously appear in endothelial cells. VEGF was cloned as a vascular endothelial cell-specific growth factor in pituitary follicular cell culture supernatant by Ferrara et al. [13].

The analysis of the nucleotide sequence of cDNA has revealed that VEGF is the same factor as vascular permeability factor (VPF) [14]. The biological activity of VEGF is specific for vascular endothelial cells, which promotes their proliferation and enhances the extravascular permeability of blood components [15]. VEGF is highly expressed in lesions of various diseases associated with angiogenesis, including cancer, and its relationship with these diseases has been attracting attention [16]. VEGF is mainly produced by cells around blood vessels, binds as a ligand to the vascular endothelial growth factor receptor (VEGFR) on the surface of vascular endothelial cells, and the receptor tyrosine kinase is activated. Signals are transmitted inside the cell, changing the function and structure of the cell. Malignant tumors also produce VEGF, drawing new vascular networks from nearby blood vessels to provide the nutrients and oxygen needed for the growth of cancer itself. Tumors without angiogenesis were reported not to grow larger than 2–3 mm^3^ [5]. In addition, new blood vessels are used for metastasis. For these reasons, angiogenesis inhibitor treatment targeting VEGF is being promoted.

Vascular stabilizer Angiopoietin-1 (Ang-1) binds to a tyrosine kinase type receptor Tie2 on the endothelial cells. It strengthens the adhesion between the endothelial cells or supporting cells (wall cells) and controls the stabilization of the vascular structure. Ang-1 is the molecule that has a coiled-coil domain and a fibrinogen-like domain and is secreted as a 70 kDa glycoprotein. On the other hand, Ang-2 exists as an antagonist of Ang-1, which binds to Tie2 but does not cause the phosphorylation of tyrosine kinase, so no signal is transmitted. Vascular maturation requires stabilizing blood vessels by interacting with Ang-1 dominant vascular endothelial cells and vascular supporting cells [17]. On the other hand, vascular retraction involves the destabilization of vascular supporting cells due to Ang-2 dominance and the cell death of vascular endothelial cells due to the lack of VEGF action (Figure 1) [4].

### 1.5. Polyphenols

Polyphenol is a general term for compounds with two or more phenolic hydroxyl groups (hydroxyl groups bonded to aromatic rings such as benzene and naphthalene ring) in the same molecule, and many types of polyphenols have been found. It is widely distributed in more than 5000 species, mainly in the plant world, and each has structural characteristics [18]. Flavonoids are a type of polyphenol synthesized by plants that have a standard structure of diphenyl propane, in which three carbons are linked to two benzene rings. Antioxidant activity is expected because it has a reducing phenol group as a structural feature. In addition, since diphenyl propane has a hydrophobic planar structure, it is expected to have a non-specific affinity to biological membranes, an antagonistic bond to a receptor using an aromatic hydrocarbon, or a bond to the active center of the enzyme as a substrate [19].

The polyphenols used in this study are shown in Appendix A. Resveratrol is a polyphenol discovered as an antioxidant with a resorcinol structure in the toxic plant *Veratrum oxyseum*. The resveratrol of red wine was reported to contribute to the prevention of heart disease. Currently, antioxidant effects that prevent cerebral infarction and arteriosclerosis are being studied [20]. Apigenin, luteolin, and quercetin are the plant flavonoids [18]. Apigenin is found in celery and parsley, and since it has no side effects, it is widely used as an anxiolytic and tranquilizer, and it has antioxidant, anti-cancer, and anti-inflammatory effects. Perilla, spring chrysanthemum, and peppers contains luteolin, which has the strongest anti-allergic and anti-inflammatory effects among flavonoids (e.g., suppressing allergic symptoms such as pollinosis and atopy). Quercetin is found in onions, apples, and green tea. Since it exhibits strong antioxidant activity, it is expected to play an essential role in preventing lifestyle-related diseases such as cancer, arteriosclerosis, and diabetes caused by oxidative stress [18].

Flavonoid compounds have a wide range of bioregulatory functions [19]. As has been reported by many epidemiological research results on flavonoid intake and disease prevention, including cancer prevention by catechins, much attention has been paid to the physiological functionality of flavonoids. Currently, they are positioned as a typical functional ingredient of plant-based foods, and research on disease prevention and health maintenance is actively underway. The primary physiological activities of flavonoids reported to date include anti-arteriosclerosis, anti-cancer, and central nervous system protective effects.

In this study, we focused on the angiogenesis-suppressing effect of antioxidant activity among the many bioactive functions of polyphenols [21,22,23]. Oxidative stress is an essential factor in angiopathy, such as arteriosclerosis. Lifestyle-related diseases such as hypertension, diabetes, and hyperlipidemia are considered to occur because of the direct inactivation of nitric oxide (NO) by the production of reactive oxygen species via NADH/NADPH oxidase [24]. It has been suggested that active oxygen exacerbates vascular endothelial dysfunction and can begin cardiovascular disease. Therefore, a method of suppressing oxidative stress is critical for preventing and treating angiopathy.

### 1.6. Molecular Pathway of Angiogenesis

The molecular markers of angiogenesis are as follows; von Willebrand factor (vWF), one of the platelet coagulation factors that forms a platelet plug; platelet endothelial cell adhesion molecule-1 (PECAM-1), which accumulates at the vascular cell adhesion site and connects endothelial cells; α-smooth muscle actin (α-SMA), which is present in the smooth muscle cells of the blood vessel wall; and Flk-1, a VEGF receptor protein-tyrosine kinase [25].

The arachidonic acid cascade is a metabolic pathway that produces prostaglandins (PGs) and thromboxane from arachidonic acid. Cyclooxygenase (COX) introduces and adds two molecules of oxygen to arachidonic acid to synthesize prostaglandin G2 (PGG2) and cleaves the hydroperoxide at the fifteenth position of PGG2 to produce prostaglandin H2 (PGH2). It is an enzyme that catalyzes the two types of reactions that are made and is one of the rate-determining enzymes for the synthesis of prostaglandins and thromboxanes. Two types of COX are known: type 1 and type 2. Type 1 (COX-1) is a constituent enzyme that is expressed in most tissues and cells. The expression of type 2 (COX-2) is induced by cytokines involved in inflammation such as interleukin-1α (IL-1α), tumor necrosis factor (TNF-α), lipopolysaccharide (LPS), and growth factors [26]. Peroxisome proliferator-activated receptor γ (PPARγ) is a member of the nuclear receptor superfamily and is activated in a ligand-gated manner to regulate target gene expression, mainly lipid metabolism, inflammation, and carcinogenesis [27]. The peroxisome proliferator response element sequence, which is the binding site for PPARγ, exists upstream of the promoter region of COX-2. The administration of ciglidazone, a thiazolidine-based PPARγ agonist, to colorectal cancer cells HT-29 reduces the expression of COX-2, suggesting that the activation of PPARγ suppresses the expression of COX-2 [28]. In addition, it has been reported that VEGF is involved in the induction of COX-2 expression in endothelial cells [29]. A potent anti-angiogenic factor, retinal pigment epithelial-derived factor (PEDF), up-regulated the transcription of PPARγ and induced apoptosis in HUVEC [30].

Nitric oxide synthase (NOS) [24] is a cellular signaling factor, and endothelial NOS (eNOS) is present in vascular endothelial cells and platelets. Elevated reactive oxygen species (ROS) in vascular tissue decrease eNOS expression. In addition, if ROS in vascular tissue decrease, then the eNOS expression increases.

### 1.7. The Aim of This Research

The control of angiogenesis is essential in disease treatment, such as angiogenesis suppression therapy in cancer, engraftment of transplanted three-dimensional tissue in regenerative medicine, and cell therapy for patients with severe ischemic disease. This study aimed to control angiogenesis, quantitatively analyze the effects of molecules that promote or suppress vascular tissue formation in culture, and clarify the mechanism of action.

## 2. Materials and Methods

### 2.1. Materials

Normal human umbilical vein endothelial cells (HUVECs) were purchased as an angiogenesis kit (Kurabo, Osaka, Japan; KZ-1000). Human rAng-1-producing 107-35 CHO cells were made by NN. DFAT-D1 cells were established from mature adipocytes of adult ddY mice [31].

Polyphenols apigenin and luteolin (Fujifilm Wako Chemicals) were dissolved in methanol, resveratrol, and quercetin (Fujifilm Wako Chemicals) dissolved in ethanol. Recombinant Ang-1 protein was purchased from R&D Systems. CD31 antibody (tube formation indicator), VEGF-A (positive control), and suramin (negative control) were purchased from Kurabo.

### 2.2. Angiogenesis Assay

HUVECs were cultured according to the manufacturer’s instructions. Briefly, HUVECs were co-cultured with normal human dermal fibroblasts at an optimal concentration in angiogenic medium-2 (Kurabo KZ-1500) on a 24-well plate. Promotive control VEGF-A (10 ng/mL), inhibitory control suramin (50 μM) or test materials such as cells or a conditioned medium or molecules were added to each well and cultured in a 5% CO_2_ incubator at 37 °C. In the presence of 10 ng/mL VEGF-A, 8.0, 4.0, 2.0, 1.0 × 10^5^ cells/mL 107-35 CHO or DFAT cells were added and directly co-cultured. Alternatively, 200, 150, 100, and 50 µL/mL of the culture supernatant of 107-35 CHO cells (DMEM + 10% FBS + 1% PS culture for 3 days) or commercially available rAng-1 800, 400, 200, and 100 ng/mL was added to the culture.

To test the effect of polyphenols and flavonoids on angiogenesis, 2.0, 1.0, 0.50, and 0.25 μM apigenin, or 20, 10, 5.0, 2.5 μM luteolin, quercetin or resveratrol were added to the wells.

### 2.3. Quantitative Analysis

After 11 days, the cells were fixed and subjected to immunohistochemical staining by using antibodies against CD31 (PECAM-1) (1:4000). After incubation at 37 °C for 1 h, alkaline phosphatase-conjugated goat anti-mouse IgG (1:500, Kurabo) was added and incubated at 37 °C for 1 h. Immunoreactivity was visualized using 5-bromo-4-chloro-3-indolyl-phosphate (BCIP) and nitro blue tetrazolium (NBT) (Vector Laboratories, Inc., Burlingame, CA, USA). Microscopic images were analyzed by Kurabo angiogenesis quantification software (KSW-5000U). Quantitative analyses were performed for 4 parameters; lumen area (area), lumen length (length), number of lumen intersections (joint), and number of paths (path) [32]. Each sample was tested in triplicate wells. Experiments were repeated more than five times. The statistical one-way analysis of variance (ANOVA) and Turkey’s multiple comparisons test was performed with Prism software. *p* values < 0.05 were considered statistically significant.

### 2.4. Gene Expression Analysis

Cells were collected from culture wells, and total RNA was extracted using the RNeasy Mini Kit (QIAGEN). Super Script™II Reverse Transcriptase (Invitrogen) and Oligo (dT) 12–18 Primer (Invitrogen) were used for cDNA synthesis. TaKaRa Emerald Amp^®^ was used for PCR. The reaction conditions were as follows, heat denaturation step 95 °C/30 s, annealing temperature 52–57 °C/30 s, extension step 72 °C/1 min, and repeated 30–35 cycles. The PCR product was electrophoresed using a 1.5–2.0% agarose gel and visualized by ultraviolet irradiation. The analyzed genes were vWF, PECAM-1, α-SMA, Flk-1 for angioplasty marker, and COX-1, COX-2, and PPARγ for arachidonic acid cascade, and eNOS for the oxidative stress-related pathway. GAPDH was used as the endogenous control gene.

## 3. Results

### 3.1. Establishment and Analysis of rAng-1-Producing CHO Cells

To generate rAng-1-producing cells, the human angiopoietin-1 (Ang-1) gene, which is involved in stabilizing vascular structure in angiogenesis, was integrated into CHO cells using the FLAG expression vector (Sigma-Aldrich, MO, USA). Among the cells that stably produce rAng-1, the 107-35 CHO cell line, which has the highest survival-promoting effect on HUVEC, was selected (Figure 2, left). As a result of examining the biochemical properties by Western blotting or immunofluorescence, this cell line’s high rAng-1 production ability was confirmed (Figure 2, middle, upper right).

Before using this cell line for the angiogenesis assay, we confirmed the amount of Ang-1 produced in the conditioned medium. Compared to 25 ng commercial rAng-1, it was revealed that 5 µL of 1 or 3d culture supernatant (conditioned medium; CM) of 107-35 CHO cells contained more than 1 µg of Ang-1 (Figure 2, lower right).

rAng-1 produced by this cell line showed a Tie-2 phosphorylation-inducing effect and a cell migration effect on bovine corneal endothelial cells (BCECs) (data not shown).

### 3.2. Promotion of Angiogenesis by rAng-1

In the presence of 10 ng/mL VEGF-A, 107-35 CHO cells, the culture supernatant of 107-35 CHO cells or commercially available rAng-1 was added to the HUVEC culture to observe its promoting effect on angiogenesis.

After culturing for 11 days, immunohistochemical staining with CD31 (PECAM-1) antibody was performed (Figure 3). CD31, a 130 kDa membrane protein belonging to the immunoglobulin superfamily and expressed in embryonic and adult endothelial cells. It plays an essential role in the interaction among vascular endothelial cells during angiogenesis. The results of immunohistochemical staining with the CD31 antibody revealed the formation of a vascular network.

Image analysis using angiogenesis software quantified four parameters: lumen area (area), lumen length (length), number of lumen intersections (joint), and number of paths (path) (Figure 4). When the control with only VEGF-A was set to 100, the score co-cultured with 107-35 CHO cells increased in a cell concentration-dependent manner, and when co-cultured with 8.0 × 10^5^ cells/mL, it was three times higher than the control. The score increased depending on the amount of conditioned medium of 107-35 CHO cells, and the 200 µL conditioned medium showed a score three times higher than that of the control. When commercially available rAng-1 was added, the score increased in a concentration-dependent manner, and when 800 ng/mL was added, the score was approximately 2.2 times that of the control. In cultures with medium alone (without VEGF-A) or with VEGF-A and suramin, the score was approximately 50% of the control. From these results, the concentration-dependent angiogenesis-promoting effect of rAng-1 was confirmed. The results of one-way ANOVA analyses (N = 3) for the individual parameters were presented as the mean ± standard deviations (SD) in Appendix A.

### 3.3. Promotion of Angiogenesis by Co-Culture with DFAT-D1 Cells

In the presence of VEGF, 8.0, 4.0, 2.0, and 1.0 × 10^5^ cells/mL DFAT-D1 was added to the HUVEC culture system. Unlike the 107-35 CHO cell co-culture, the linear concentration-dependent effect was not seen. However, compared to the VEGF-A-only control, the scores for the four parameters were approximately two times higher on average. These results suggest that the co-culture of DFAT-D1 cells promotes angiogenesis (Figure 3 and Figure 4). The results of one-way ANOVA analyses (N = 3) for the individual parameters are presented as mean ± standard deviations (SD) in Appendix A.

### 3.4. Angiogenesis Inhibitory Effect of Polyphenols

Polyphenols were added to the above systems to observe the inhibitory effect on angiogenesis. Suramin was used as the inhibitory control.

After culturing for 11 days, image analysis using angiogenesis software was performed from immunohistochemically stained images (Figure 5) with CD31 antibody. The four parameters were quantified, and the control to which VEGF-A was added was set to 100 and quantified (Figure 6). As a result, polyphenols showed a dose-dependent inhibitory effect on angiogenesis. The inhibition of angiogenesis by luteolin and quercetin was particularly pronounced, with 20 µM addition suppressing the score to 10–20% of the VEGF-A-only control. Apigenin and resveratrol also showed the concentration-dependent inhibition of angiogenesis. Here, 2 µM apigenin and 20 µM resveratrol induced a reduction in the score to 40–50%. The score was approximately 20% or less in the medium alone (without VEGF-A) and approximately 30% in suramin with VEGF (Figure 6). The results of one-way ANOVA analyses (N = 3) for the individual parameters are presented as the mean ± standard deviations (SD) in Appendix A.

### 3.5. Investigation of the Molecular Mechanism by Gene Expression Analysis

After culturing for 11 days, cells were collected from individual wells adding soluble factors and total RNA was extracted and purified. Gene expression analysis was performed by RT-PCR.

We investigated the expression of markers that are indicators of angiogenesis (Figure 7); von Willebrand factor (vWF), platelet endothelial cell adhesion molecule-1 (PECAM-1), α-smooth muscle actin (α-SMA) and protein-tyrosine kinase receptor Flk-1, a receptor for VEGF. All of these are expressed in vascular endothelial cells [25].

The amount of angiogenesis-promoting factor that showed the maximum effect in Figure 3 and Figure 4 (800 ng/mL of rAng-1, 200 µL of the conditioned medium of 107-35 CHO) was added in the culture (Figure 7, top). The conditioned medium of 107-35 CHO cells (Cm) showed a remarkable increase in the expression of vWF and Flk-1 compared to the control VEGF alone (V) or medium-only control (N) without VEGF-A. PECAM-1 expression was slightly increased. On the other hand, the expression of α-SMA was already high in the control, and the expression was not changed by adding an angiogenesis-promoting factor. No significant increase in expression was observed when rAng-1 (Rec) was added.

On the other hand, when the angiogenesis inhibitor polyphenols were added (Figure 7, bottom), the expression of vWF and PECAM-1 was lower than that of the control VEGF alone (V). Apigenin (AV) and luteolin (LV) markedly suppressed α-SMA and Flk-1 expression, whereas resveratrol (RV) and quercetin (QV) showed no suppressing effect for these genes compared with VEGF alone (V). Negative control suramin (SV) repressed all four genes.

Furthermore, we analyzed the expression of COX-1, COX-2, and PPARγ, which are arachidonic acid cascade-related genes. The amount of angiogenesis-suppressing factor that showed the most significant effect in Figure 5 and Figure 6 (2 µM of apigenin, 20 µM of resveratrol, luteolin, and quercetin) were added to the culture. Resveratrol (RV) and quercetin (QV) enhanced PPARγ expression, and QV increased COX-2 expression (Figure 8, top). Luteolin (LV) repressed COX-1 (Figure 8, top). The expression of eNOS, an oxidative stress-related gene, was slightly increased by LV (Figure 8, bottom). These results suggested that polyphenol decreased ROS.

## 4. Discussion

This study analyzed the effects of molecules that quantitatively promote or suppress angiogenesis and clarified the mechanism of their action for the control of angiogenesis in disease and regenerative medicine.

### 4.1. Angiogenesis-Promoting Effect of Ang-1

We established the cell line 107-35CHO which produces the vascular stabilizer Ang-1. The conditioned medium of this cell line contained considerable amounts of recombinant proteins. A concentration-dependent angiogenesis-promoting effect was observed by co-culturing these cells with HUVEC. When the conditioned medium of this cell line or commercially available rAng-1 was added to the culture, a concentration-dependent angiogenesis-promoting effect was also shown.

Ang-2 dissociates the blood vessel wall from the endothelial cells and promoted neovascularization, whereas Ang-1, which is necessary for stabilizing neovascular tissue. Both Ang-1 and 2 binds to a receptor called Tie-2 and controls angiogenesis. Angiogenesis in culture is indicated by the linear connection of vascular endothelial cells and the formation of lumens. Like VEGF, Ang-1 is produced in cells around blood vessels and acts by binding to Tie-2 in a paracrine manner. Ang-1 promotes cell migration or suppresses apoptosis (cell death). It has been suggested that when a signal input comes from Tie-2, it acts on endothelial cells to promote angiogenic migration and aggregate pericytes. The knockout of the Ang-1 gene resulted in fetal lethality due to blood vessels’ deficiency. Conditional gene targeting with the Cre-lox system revealed that Ang-1 is essential for regulating the number and diameter of developing blood vessels but is not critical for recruiting pericytes [32].

On the other hand, platelet-derived growth factor (PDGF-B) is involved in the proliferation and migration of vascular endothelial cells and pericytes. A knockout study suggested that PDGF-B retention is essential for the proper recruitment and organization of pericytes and renal and retinal function in adult mice [33].

### 4.2. Angiogenesis Promoting Effect of DFAT

The direct co-culture of DFAT-D1 cells with HUVEC promoted angiogenesis compared to VEGF-A-only controls. Though the linear concentration-dependency was not seen, the scores for the four parameters of angiogenesis were approximately two times higher on average.

DFAT-D1 cells are preadipocyte lineage cells established from adult mouse adipocytes [31] and have a fibroblast-like morphology, however, they could re-differentiate in vitro into mature adipocytes after over 20 passages. Mouse DFAT cells established from adipose tissues [12] could differentiate into adipocyte, osteoblasts, chondrocytes. Flow cytometric analysis has revealed that Sca1, CD29, CD105, CD106 are positive, and CD11b, CD45, CD73 are negative in mouse DFAT cells [12]. The transplantation of DFAT cells (1 × 10^5^ cells/body) into a hindlimb ischemia model of SCID mice showed improved blood flow and increased mature blood vessel density [12]. DFAT cells secrete angiogenic factors VEGF-A and HGF, and their levels in the culture supernatant were measured using ELISA. It was revealed that under hypoxic conditions, the secretion of VEGF-A or HGF was significantly higher than under normoxic conditions [12].

The gene expression was analyzed using the indirect co-culture system of DFAT with vascular endothelial (mCherry-labelled Mile Seven 1; MS1) cells [12]. HGF, FGF-2, and Ang-1 expression in DFAT cells were significantly higher in the co-culture with MS1 than that in the control (DFAT alone). In contrast, the expression of VEGF-A did not change. The expression of these angiogenic factors from MS1 was considerably lower than that of the DFAT cells. TGF-β was expressed in both DFAT cells and MS1, while PDGF-BB was expressed in MS1 but not in DFAT cells. These data suggest that the co-culture of DFAT cells with endothelial cells promoted the expression of several angiogenic factors in each type of cell [12].

The collagen beads assay was applied to show the tubular structure formation. Watanabe et al. [12] performed the angiogenesis experiment by adhering DFAT cells and vascular endothelial cells to the surface of the dextran beads, embedded and cultured them in collagen gel, and monitored tube formation from the beads by tube length and area. A quantitative analysis revealed that the length and area of the tubules in the MS1 + DFAT group were significantly higher than that for the MS1 group on day 7. DFAT cell-conditioned media prepared from both normoxic and hypoxic culture conditions significantly promoted tube formation in HUVECs [12].

DFAT cells also differentiated themselves and expressed pericyte markers such as ASMA and NG2. GFP-labelled DFAT cells adhered to the outer surface of the tubules of the MS1 cells [12]. Still, it is thought that the secretion of cytokines from DFAT cells mainly promotes the tube formation of vascular endothelial cells. Human adipose-derived stem/stromal cells (ASCs) could differentiate into adipocyte, osteoblasts, chondrocytes, and smooth muscle-like cells [34]. The conditioned medium of ASCs (ASC-CM) enhanced HUVEC tube formation (Kurabo), and this response was not influenced by donor age. Considerable levels of VEGF-A and HGF were detected in ASC-CM measured with ELISA [34].

### 4.3. Angiogenesis Suppressing Effect of Polyphenol

From the results of quantitative analysis, polyphenols showed a dose-dependent inhibitory effect on angiogenesis. In particular, the inhibition by luteolin and quercetin was remarkable. Luteolin had strong anti-allergic and anti-inflammatory effects, and quercetin exhibited strong antioxidant activity. They seemed to play an essential role in suppressing oxidative stress, which is critical for the prevention and treatment of angiopathy.

Previous studies reported that antioxidant polyphenols suppressed angiogenesis. Silymarin, a natural flavonoid antioxidant, had an inhibitory effect on VEGF secretion by cancer epithelial cells and was an effective angiogenesis inhibitor on human umbilical vein endothelial cells (HUVECs) [35]. Like Watanabe et al. [12], we examined the effects of polyphenols using the collagen beads assay and confirmed that the polyphenols suppressed tube formation (data not shown).

### 4.4. Molecular Mechanisms

To clarify the molecular mechanism of these angiogenesis-promoting and -suppressing effects, we performed gene expression analysis by RT–PCR.

First, the expression of a marker that is characteristic of vascular tissue formation was examined. When a conditioned medium of rAng-1 producing 107-35CHO cells was added, the expression of vWF and Flk-1, PECAM-1 was particularly increased. Flk-1 has been reported as a marker for ES cell-derived vascular stem cells [36]. On the other hand, when an angiogenesis inhibitor polyphenols were added, the expression of vWF and PECAM-1 was suppressed compared to control VEGF alone. The suppression of α-SMA and Flk1 by apigenin and luteolin was particularly remarkable.

Polyphenols absorb free radical electron pairs and have various actions, such as inhibiting enzymes in intracellular signal transduction. To clarify how polyphenols are involved in the intracellular signal transduction of vascular endothelial cells from VEGF and suppress angiogenesis, we performed gene expression analysis focused on PPARγ, COX-1, COX-2, and eNOS. In the quantitative analysis of angiogenesis, inhibition by luteolin and quercetin was remarkable, but in the gene expression analysis, PPARγ expression was enhanced by resveratrol and quercetin. However, the expression of COX-2 increased, especially with the addition of quercetin. It has been suggested that the activation of PPARγ suppresses COX-2 expression [28]. Though in the case of resveratrol, COX-2 expression was slightly reduced but was promoted in the case of quercetin. In addition, luteolin significantly suppressed the expression of COX-1. Regarding the expression of eNOS, an oxidative stress-related gene, there was no significant difference compared with the control VEGF alone, but luteolin showed slight upregulation. These results suggested that polyphenols reduce the active oxygen ROS in vascular tissue [24].

It has been reported that polyphenols help prevent the aging of the brain due to their antioxidant and anti-inflammatory properties. The SIRT1 gene product is an NAD + -dependent deacetylase localized in the nucleus and cytoplasm of cells. It is involved in bioregulation that prolongs lifespan and delays aging by deacetylating histones, transcription factors, and enzymes and controlling their activity. SIRT1 is highly expressed in the vasculature during growth and promotes endothelial cell angiogenesis. It has been reported that resveratrol, found in grape skin and red wine, enhances the enzymatic activity of SIRT1 [37,38]. Our findings contradicted this result because resveratrol suppressed angiogenesis. We will investigate the expression analysis of SIRT1 in a future experiment.

### 4.5. Future Perspective

Recently, methods for detecting, controlling, treating, or preventing angiogenesis for diseases such as cancer and allergic diseases have been investigated [39,40,41,42,43]. We are conducting clinical trials in patients with severe lower limb ischemic disease by locally transplanting DFAT cells established by culture from patients to promote blood vessel regeneration. DFAT secreted angiogenesis factors, and indirect co-culture confirmed the angiogenesis-promoting effect [12]. Since there was a report that DFAT cells themselves differentiate into endothelial cells [44], we would like to consider a method of transplanting created blood vessels in vitro.

On the other hand, another process called arteriogenesis is known in neovascularization in a broad sense. It refers to the anatomic transformation of preexisting arterioles with increased lumen area and wall thickness and vasomotor capacities. Arteriogenesis differs from angiogenesis in several aspects, the most important being the dependence of angiogenesis on hypoxia and the dependence of arteriogenesis on inflammation [45]. We are currently investigating the effect of DFAT on arteriogenesis.

In addition, the symptoms of COVID-19, which is currently a global threat, extend beyond the lungs to affect the cardiovascular, nervous, and other organ systems. Current treatments are for short-term symptoms. The main targets are preventing the binding of viruses to receptors and cell invasion and suppressing immune runaway. Long-term consequences such as pulmonary fibrosis, demyelination, and ischemic organ damage are not addressed. Cell therapy is thought to offer great potential for treating severe COVID-19 symptoms due to its customizability and regenerative function [46]. Currently, preclinical and clinical cell therapy trials are under way regarding the immunomodulatory ability of mesenchymal stem cells for COVID-19-related lung injury treatment and the mitigation of hyperimmune response. Cell therapy for the cardiovascular system is promising, and future treatment development and elucidation of the mechanism are expected. Based on the knowledge gained from this research, we would like to take on that challenge.

## 5. Conclusions

This study aimed to quantitatively analyze the effects of molecules that promote or suppress angiogenesis and clarify the mechanism for its regulation. Based on the results of numerical image analyses, the effects of Ang-1 and polyphenols on vascular tissue formation were examined. As a result, we could clarify the concentration-dependent angiogenesis-promoting effect of Ang-1 and the concentration-dependent angiogenesis-suppressing effect of four types of polyphenols. On the other hand, a co-culture with DFAT cells also showed an angiogenesis-promoting effect. We would like to apply the control of angiogenesis by the factors clarified in this study and apply it in regenerative vascular medicine.

## 6. Patents

WO 2016158670: COMPOSITION FOR VASCULAR REGENERATION THERAPY, CONTAINING DEDIFFERENTIATED FAT CELLS AS ACTIVE INGREDIENT

## Figures and Tables

**Figure 1 biology-10-01212-f001:**
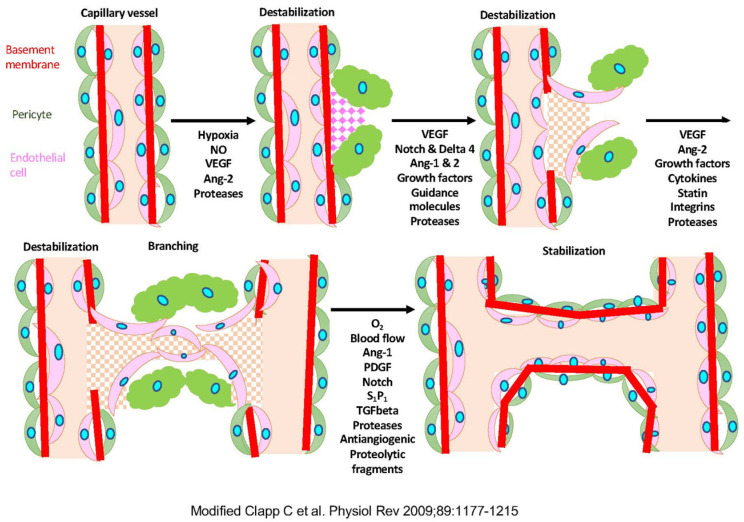
The mechanism of angiogenesis. Modified Clap C et al., 2009 [4]. Hypoxia induces the production of nitric oxide (NO) and the expression of vascular endothelial growth factor (VEGF) and angiopoietin-1, and -2 (Ang-1 and Ang-2). The extracellular matrix (ECM) proteases increase the permeability of the capillary vessel wall, and endothelial cells migrate and proliferate to form tubules. PDGF, platelet-derived growth factor; S1P1, sphingosine-1-phosphate-1; TGFbeta, transforming growth factor-beta.

**Figure 2 biology-10-01212-f002:**
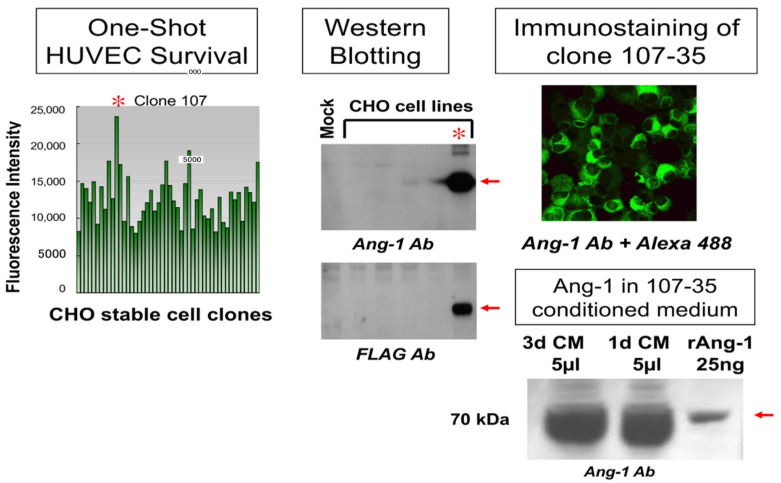
Screening of rAng1-producing cells. The human angiopoietin-1 (Ang-1) gene was integrated into CHO cells using the FLAG expression vector. The 107-35 CHO cell line (*) has the highest survival-promoting effect on HUVEC (**left**). Western blotting or immunofluorescence showed the rAng-1 production ability (**middle**, upper right). Five microliters (5 µL) of 1 or 3d culture supernatant (CM) contained more than 1 µg of Ang-1 (lower **right**).

**Figure 3 biology-10-01212-f003:**
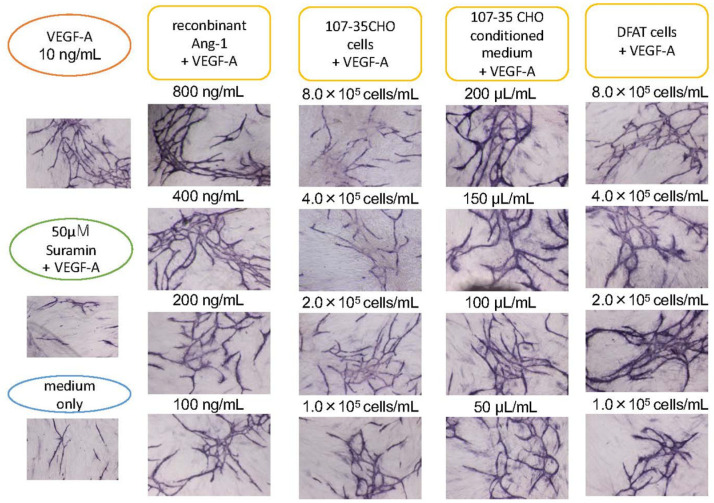
Promoting effect of Ang-1 on angiogenesis. HUVECs were cultured with various concentrations of 107-35CHO cells, conditioned medium, and rAng-1. In the presence of 10 ng/mL VEGF-A, 8.0, 4.0, 2.0, and 1.0 × 10^5^ cells/mL 107-35 CHO cells were added to HUVEC into a 24-well culture. Alternatively, 200, 150, 100, and 50 µL/mL of the culture supernatant of 107-35 CHO cells or commercially available rAng-1 800, 400, 200, and 100 ng/mL was added to the culture. rAng-1 showed remarkable effects. A co-culture with 8.0, 4.0, 2.0, and 1.0 × 10^5^ cells/mL DFAT cell promoted angiogenesis compared to control.

**Figure 4 biology-10-01212-f004:**
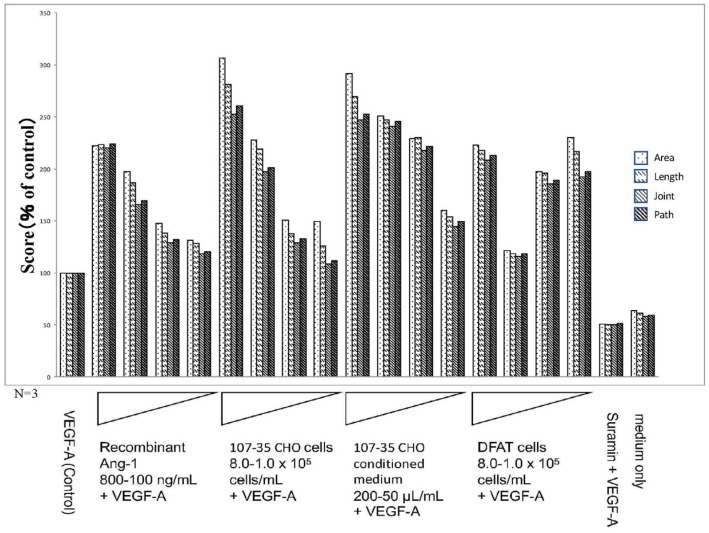
Results of the quantitative analysis by using the Kurabo Angiogenesis Image Analyzer (promotion). The stained images were subjected to the quantitative microscope analysis for 4 parameters: lumen area (area), lumen length (length), number of lumen intersections (joint), and number of paths (path) using the Kurabo angiogenesis quantification software. In the presence of 10 ng/mL VEGF-A (Control), the factors added to each culture well were rAng-1 (recombinant Ang-1; 800–100 ng/mL), CHO (rAng-1-producing 107-35 CHO cells; 8.0–1.0 × 10^5^ cells/mL), CHO-CM (107-35 CHO cell-conditioned medium; 200–50 μL/mL), DFAT (DFAT cells; 8.0–1.0 × 10^5^ cells/mL), and suramin (50 μM). Medium only contained no VEGF-A. The results of quantitative analyses suggested that Ang-1 has a promoting effect on angiogenesis.

**Figure 5 biology-10-01212-f005:**
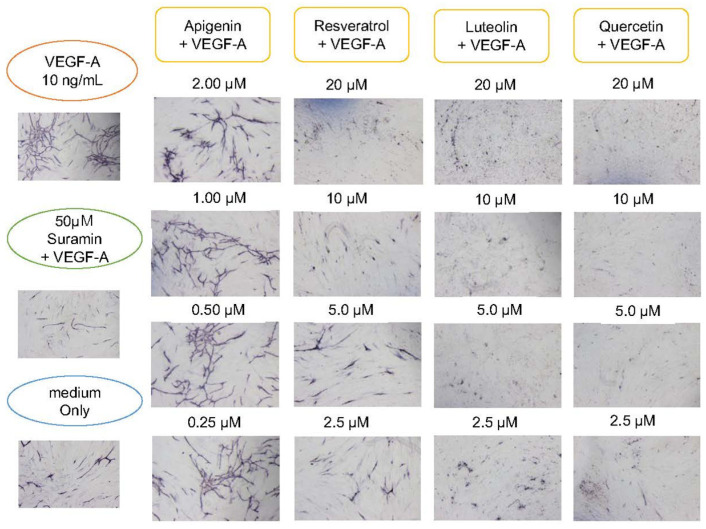
Suppressing effect of polyphenols on angiogenesis. HUVECs were cultured with various concentrations of polyphenols or flavonoids. In the presence of 10 ng/mL VEGF-A, 2.0, 1.0, 0.50, and 0.25 μM apigenin, or 20, 10, 5.0, and 2.5 μM resveratrol, luteolin, quercetin and negative control suramin (50 μM) were added to HUVEC in a 24-well culture. Medium only contained no VEGF-A. After 11 days, cells were fixed and subjected to immunohistochemical staining by using antibodies against CD31. Flavonoids or polyphenol showed inhibitory effects on angiogenesis in a dose-dependent manner. Particularly, luteolin and quercetin showed remarkable effects of suppressing angiogenesis.

**Figure 6 biology-10-01212-f006:**
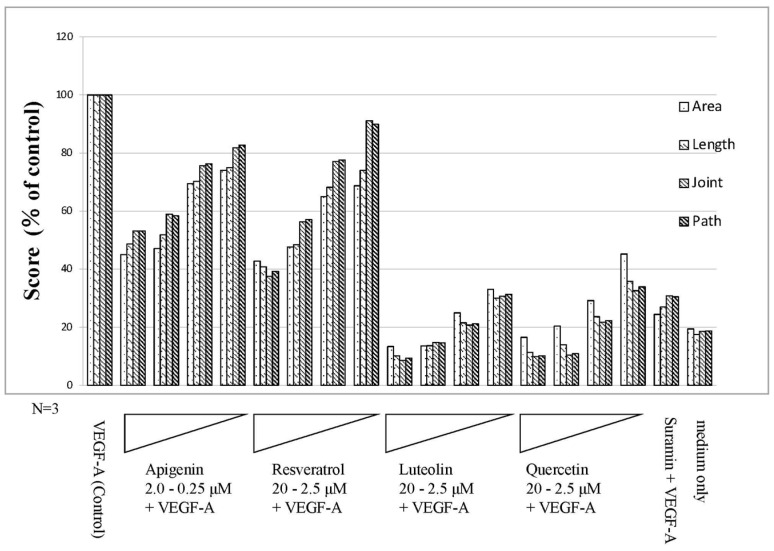
Results of quantitative analysis using the Kurabo Angiogenesis Image Analyzer (Polyphenols). The stained images were subjected to the quantitative microscope analysis for 4 parameters: lumen area (area), lumen length (length), number of lumen intersections (joint), and number of paths (path) by using the Kurabo angiogenesis quantification software. In the presence of 10 ng/mL VEGF-A (control), In the presence of 10 ng/mL VEGF-A, 2.0–0.25 μM apigenin, or 20–2.5 μM resveratrol, luteolin, quercetin, and negative control suramin (50 μM) were added to HUVEC in a 24-well culture. Medium only contained no VEGF-A. Flavonoids or polyphenol showed inhibitory effects on angiogenesis in a dose-dependent manner. Notably, luteolin and quercetin showed remarkable effects in terms of suppressing angiogenesis.

**Figure 7 biology-10-01212-f007:**
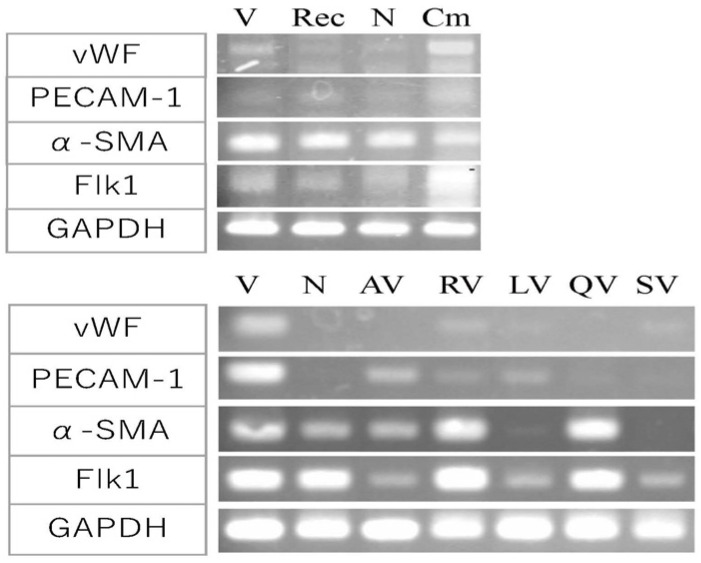
Effect of promotion and suppression factor on gene expression in angiogenesis. Gene expression analysis for indicators of angiogenesis; vWF, PECAM-1, α-SMA, Flk-1, and GAPDH as a control. The effects of angiogenesis-promoting factors (top). V, VEGF-A; Rec, recombinant Ang-1 + VEGF-A; N, medium only; Cm, 107-35 CHO conditioned medium + VEGF-A. Cm induced a remarkable increase in the expression of vWF and Flk-1 compared to the control V or N. The effects of angiogenesis-suppressing factors (bottom). V, VEGF-A; N, medium only; AV, apigenin + VEGF-A; RV, resveratrol + VEGF-A; LV, luteolin + VEGF; QV, quercetin + VEGF-A; SV, suramin + VEGF-A. The expression of vWF and PECAM-1 were suppressed by polyphenols. AV and LV markedly suppressed α-SMA and Flk-1 expression.

**Figure 8 biology-10-01212-f008:**
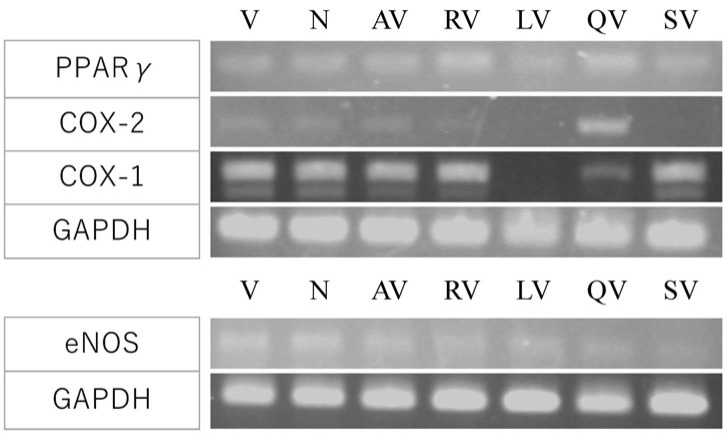
Effect of polyphenols on COX pathway and eNOS gene expression. The expression of arachidonic acid cascade-related genes; COX-1, COX-2, and PPAR γ (top), and eNOS, an oxidative stress-related gene (bottom) were analyzed. V, VEGF-A; N, medium only; AV, apigenin + VEGF-A; RV, resveratrol + VEGF-A; LV, luteolin + VEGF; QV, quercetin + VEGF-A; SV, suramin + VEGF-A. RV and QV enhanced PPARγ expression. QV upregulated COX-2 and LV repressed COX-1. eNOS expression was slightly enhanced by LV.

## Data Availability

The data presented in this study are available on request from the corresponding author.

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
