# Peer review of "Quantitative Analysis of Factors Regulating Angiogenesis for Stem Cell Therapy"

_biology, 2021, doi:10.3390/biology10111212_

Round 1
Reviewer 1 Report
The manuscript entitled “Quantitative Analysis of Factors Regulating Angiogenesis for 2 Stem Cell Therapy” reports an interesting effects of Ang-1 and DFAT in promoting angiogenesis, and the suppressing effect of flavonoids on it. Although testing the potential factors in regulating in-vitro model of angiogenesis is well described using a sound methodology and data analysis, its reproducibility and the number of samples and experiments need to be clarified further. In addition, the limitations of the in-vitro model and how it will be tested in the future model were not discussed. Addressing those issues in the discussion will further strengthen the manuscript.
Author Response
Dear Reviewer 1,
Thank you very much for your careful reading and kind suggestions.
The uploaded manuscript is a revised version of our research article,
Title: Quantitative Analysis of Factors Regulating Angiogenesis for Stem Cell Therapy
Authors: Takahiro Shimazaki, Nobuhiro Noro, Kazuhiro Hagikura, Taro Matsumoto, Chikako Yoshida-Noro *
Submitted to the section: Cardiovascular Biology, Stem Cells for Cardiovascular Biology and Medicine
One version is highlighted with red letters for revision and blue letters for English edit so that the editors and reviewers can easily view any changes. The other version is the final one in all black letters.
According to your suggestions, we revised the paper. Our responses are as follows, shown in red letters.
Moderate English changes required
We did English editing through MDPI.
Are the results clearly presented? Can be improved.
Are the conclusions supported by the results? Can be improved.
We revised the description in the results and discussion.
Comments to Biology-1387408
The manuscript entitled “Quantitative Analysis of Factors Regulating Angiogenesis for Stem Cell Therapy” reports an interesting effect of Ang-1 and DFAT in promoting angiogenesis, and the suppressing effect of flavonoids on it. Although testing the potential factors in regulating in-vitro model of angiogenesis is well described using a sound methodology and data analysis, its reproducibility and the number of samples and experiments need to be clarified further.
We performed the same experiments more than five times and each sample was triplicated as shown in Figures S2 and S3. We added the description in Materials and Methods.
In addition, the limitations of the in-vitro model and how it will be tested in the future model were not discussed. Addressing those issues in the discussion will further strengthen the manuscript.
In the discussion section, we added some explanations about in vivo experiments done by Watanabe et al. [12].
We hope the revised version will be suitable for publication and look forward to hearing from you regarding our submission.
Thank you in advance for your kind consideration of this paper.
Sincerely yours,
Chikako Yoshida-Noro, Ph.D
Division of Cell Regeneration and Transplantation, Department of Functional Morphology, Nihon University School of Medicine, 173-8610 Tokyo, Japan
Reviewer 2 Report
Comments to Biology-1387408
Data were so interesting. However, some details were missing in the methods, and some experimental designs were inappropriate to assure the authors’ statements.
Line209
“Is induced and expressed [26].” It does not make sense.
Line240
“VEGA-A” is a typo of “VEGF-A”?
Line243
How many HUVEC cells were cultured in each experiment? Additionally, the authors should explain more about the co-culturing method and the way of collecting conditioned medium of 107-35 CO cells. Furthermore, half-width spaces are missing throughout the manuscript. Half-width space should be inserted before the unit of the concentration of protein.
Line276
The abbreviation of angiopoietin-1 is determined as ‘Ang1’ in Introduction.
Figure 1
The authors should quantify the amount of Ang1 produced by 107-35 CHO cells. It seems to be picogram-order and smaller than the volume of recombinant protein that the authors used in HUVEC cultures in Figure 3.
Line299
Immunofluoresence staining is insufficient for clarification of the formation of a vascular network. Additional experiment, such as a tube formation assay, will be needed.
Line304, 314
The authors should use superscript letters.
Line302-312
The description about the results is improper. For example, in line 304, the score from co-culture with 8.0 × 105 107-35 CHO cells is not 3 times higher than control.
Line317
The authors should describe the precise condition of cells in this result. The statement of “the scores for the four parameters were up to 2.3 times higher” might be incorrect.
Figure 4
Although the authors described 100-800 ng/ml rAng1 was used for HUVEC culture in the Materials and Methods section, 400-800 ng/ml recombinant Ang1 was used in this experiment. Please explain about this or correct the sentences.
Line344-345
The score of 20 μM quercetin was 10-20% of the control but that of resveratrol was not. In addition, the score of 20 μM luteolin was not 40-50% of the control. The authors must describe the results more carefully. The graphs of Fig.4 and Fig.6 are illegible. Please modify them to clarify each culture conditions.
Line364
Please state which cells were used for RT-PCR.
Line371
α-SMA is not a cell. Please correct the sentence.
Line382
“The amount of angiogenesis-promoting factor…” Please describe the concentration of them.
Line383
Please avoid placing a numeral at the beginning of a sentence.
Throughout the results
Statistical analysis is completely absent.
Discussion
The Discussion section is poorly structured. The authors did not discuss about the results.
Author Response
Dear Reviewer 2,
Thank you very much for your careful reading and kind suggestions.
Uploaded manuscript is a revised version of our research article,
Title: Quantitative Analysis of Factors Regulating Angiogenesis for Stem Cell Therapy
Authors: Takahiro Shimazaki, Nobuhiro Noro, Kazuhiro Hagikura, Taro Matsumoto, Chikako Yoshida-Noro *
Submitted to the section: Cardiovascular Biology, Stem Cells for Cardiovascular Biology and Medicine
One version is highlighted with red letters for revision and blue letters for English edit so that the editors and reviewers can easily view any changes. The other version is the final one in all black letters.
According to your suggestions, we revised the paper. Our responses are as follows, shown in red letters.
English language and style are acceptable/minor spell check required
We did English editing through MDPI.
Does the introduction provide sufficient background and include all relevant references?
Must be improved.
We re-organized the introduction for a better understanding of the background and also in discussion with a new reference.
Is the research design appropriate? Can be improved.
Are the methods adequately described? Not applicable.
Are the results clearly presented? Not applicable.
Are the conclusions supported by the results? Must be improved.
We revised and re-organized the description in the methods, results, discussion, and conclusions to understand the research design better.
Comments to Biology-1387408
The data were so interesting. However, some details were missing in the methods, and some experimental designs were inappropriate to assure the authors' statements.
Line209
"Is induced and expressed [26]." It does not make sense.
We corrected it.
Line240
"VEGA-A" is a typo of "VEGF-A"?
We corrected it.
Line243
How many HUVEC cells were cultured in each experiment? Additionally, the authors should explain more about the co-culturing method and collecting conditioned medium of 107-35 CO cells. Furthermore, half-width spaces are missing throughout the manuscript.
We did the experiments according to the manufacturer's instructions. Briefly, HUVECs were co-cultured with normal human dermal fibroblasts at an optimal concentration in angiogenic medium-2 (Kurabo KZ-1500) on a 24-well plate. It might contain some matrix protein. But the instruction does not show the number of cells or details of a matrix as it is the manufacturer's secret. We added some detailed descriptions of co-culture.
Half-width space should be inserted before the unit of the concentration of protein.
We inserted half-width space before the unit of the concentration of protein.
Line276
The abbreviation of angiopoietin-1 is determined as 'Ang1' in the Introduction.
We corrected the description in the introduction to Ang-1.
Figure 1
The authors should quantify the amount of Ang1 produced by 107-35 CHO cells. It seems to be picogram-order and smaller than the volume of a recombinant protein that the authors used in HUVEC cultures in Figure 3.
We added the electrophoresis data in Figure 2 to show the amount of Ang-1 produced by 107-35 CHO cells compared to rAng-1 protein.
Line299
Immunostaining is insufficient for clarification of the formation of a vascular network. Additional experiments, such as a tube formation assay, will be needed.
As mentioned in the discussion, Watanabe et al. [12] performed the angiogenesis experiment by adhering DFAT and vascular endothelial cells on the surface of the dextran beads, embedded them in collagen gel and cultured, and monitored tube formation from the beads by tube length and area. We also examined the effects of flavonoids using this method and confirmed that the flavonoids suppressed tube formation (data not shown).
Line304, 314
The authors should use superscript letters.
We corrected them to superscript letters.
Line302-312
The description about the results is improper. For example, in line 304, the score from co-culture with 8.0 × 105 107-35 CHO cells is not three times higher than control.
We changed the figure 4 to the same data as FigureS2. It showed the appropriate results.
Line317
The authors should describe the precise condition of cells in this result. The statement of "the scores for the four parameters were up to 2.3 times higher" might be incorrect.
We changed Figure 4 to the same data as FigureS2. It showed the appropriate results.
Figure 4
Although the authors described 100-800 ng/ml rAng1 was used for HUVEC culture in the Materials and Methods section, 400-800 ng/ml recombinant Ang1 was used in this experiment. Please explain about this or correct the sentences.
We corrected it to 100-800 ng/ml rAng-1.
Line344-345
The score of 20 μM quercetin was 10-20% of the control, but that of resveratrol was not. In addition, the score of 20 μM luteolin was not 40-50% of the control. The authors must describe the results more carefully. The graphs of Fig.4 and Fig.6 are illegible. Please modify them to clarify each culture condition.
We changed the Figure 4 and 6 to the same data as FigureS2 and S3. It showed the appropriate results. I have corrected the mistake in the description between quercetin and resveratrol.
Line364
Please state which cells were used for RT-PCR.
Cells were collected from individual wells of 24 well-plate. We did not use direct co-culture with DFAT for gene expression analysis of HUVEC as the indirect co-culture data was shown in Watanabe et al. [12]. They showed that DFAT cells express pericyte markers in the indirect co-culture with endothelial cells.
Line371
α-SMA is not a cell. Please correct the sentence.
We corrected it.
Line382
"The amount of angiogenesis-promoting factor…" Please describe the concentration of them.
We described the concentration.
Line383
Please avoid placing a numeral at the beginning of a sentence.
We changed.
Throughout the results
Statistical analysis is completely absent.
We have done statistical analysis with Angiogenesis Kit Software as shown in Figures 1S and 2S.
Discussion
The Discussion section is poorly structured. The authors did not discuss about the results.
We added some descriptions and re-organized the discussion section.
We hope the revised version will be suitable for publication and look forward to hearing from you regarding our submission.
Thank you in advance for your kind consideration of this paper.
Sincerely yours,
Chikako Yoshida-Noro, Ph.D
Division of Cell Regeneration and Transplantation, Department of Functional Morphology, Nihon University S
Reviewer 3 Report
What is the HUVEC cell density are used for the assays? The authors have not provided it.
Methods section is not clear and little bit confusing. The authors explained part of methods in the results section. It would be more comprehensible if complete methods section is at one place.
All angiogenesis tube formation assay requires using of somekind of basement membrane matrix such as Matrigel or Geltrex. It wasn’t mentioned if any basement membrane matrix was used in the angiogenesis.
The angiogenic gene expression data of DFAT and HUVEC co-culture was not shown.
Figure 7: Why there is discrepancy gene expression between V samples in promoting and suppressing angiogenesis? The V (VEGF-A) only samples are essentially same in both conditions.
Why are the suppression angiogenesis experiments were not conducted with DFAT co-culture?
DFAT cells may have secreted many different types of secretions. The authors have not studied what secretions are released from DFAT cells but imply that Ang-1 is secreted. The enhanced angiogenic effect of HUVECs in DFAT co-culture might be a result of several factors rather than just Ang-1.
There are several typos throughout the manuscript.
Author Response
Dear Reviewer 3,
Thank you very much for your careful reading and kind suggestions.
Uploaded manuscript is a revised version of our research article,
Title: Quantitative Analysis of Factors Regulating Angiogenesis for Stem Cell Therapy
Authors: Takahiro Shimazaki, Nobuhiro Noro, Kazuhiro Hagikura, Taro Matsumoto, Chikako Yoshida-Noro *
Submitted to the section: Cardiovascular Biology, Stem Cells for Cardiovascular Biology and Medicine
One version is highlighted with red letters for revision and blue letters for English edit so that the editors and reviewers can easily view any changes. The other version is the final one in all black letters.
According to your suggestions, we revised the paper. Our responses are as follows, shown in red letters.
Moderate English changes required
We did English editing through MDPI.
Is the research design appropriate? Can be improved.
Are the methods adequately described? Must be improved.
Are the results clearly presented? Can be improved.
Are the conclusions supported by the results? Must be improved.
We revised and re-organized the description in the methods, results, discussion, and conclusions to understand the research design better.
Comments to Biology-1387408
What is the HUVEC cell density are used for the assays? The authors have not provided it.
Methods section is not clear and little bit confusing. The authors explained part of methods in the results section. It would be more comprehensible if complete methods section is at one place.
All angiogenesis tube formation assay requires using of some kind of basement membrane matrix such as Matrigel or Geltrex. It wasn't mentioned if any basement membrane matrix was used in the angiogenesis.
We did the experiments according to the manufacturer's instruction. Briefly, HUVECs were co-cultured with normal human dermal fibroblasts at an optimal concentration in angiogenic medium-2 (Kurabo KZ-1500) on a 24-well plate. It might contain some matrix protein. But the instruction does not show the number of cells or details of the matrix as it is the manufacturer's secret. We added some detailed descriptions of co-culture. We changed some of the descriptions in the result section moved to the method section.
The angiogenic gene expression data of DFAT and HUVEC co-culture was not shown.
As gene expression was analyzed for the whole cells collected from the wells, we did not use co-culture samples in this paper. Please refer to the Watanabe et al., [12]. They showed that DFAT cells express pericyte markers in the co-culture with endothelial cells.
Figure 7: Why there is a discrepancy gene expression between V samples in promoting and suppressing angiogenesis? The V (VEGF-A) only samples are essentially same in both conditions.
They were tested separately, so there were some errors, but the trends were the same.
Why are the suppression angiogenesis experiments were not conducted with DFAT co-culture?
As the results in Watanabe et al., [12], we knew that DFAT co-culture promoted angiogenesis. So we did not use DFAT co-culture in the suppression experiments.
DFAT cells may have secreted many different types of secretions. The authors have not studied what secretions are released from DFAT cells but imply that Ang-1 is secreted. The enhanced angiogenic effect of HUVECs in DFAT co-culture might be a result of several factors rather than just Ang-1.
Please refer to the Watanabe et al., [12]. They showed that DFAT secreted many factors related to angiogenesis by using an indirect co-culture system. In this paper, we addressed Ang-1 for promoting factor and polyphenols for suppressing factors.
There are several typos throughout the manuscript.
We corrected them by English edit.
We hope the revised version will be suitable for publication and look forward to hearing from you regarding our submission.
Thank you in advance for your kind consideration of this paper.
Sincerely yours,
Chikako Yoshida-Noro, Ph.D
Division of Cell Regeneration and Transplantation, Department of Functional Morphology, Nihon University School of Medicine, 173-8610 Tokyo, Japan
Round 2
Reviewer 2 Report
The manuscript has been improved and spelling errors were corrected.
Further clarification of the angiogenic effects of DFAT and the precise signaling mechanism will help for researchers and clinicians to translate them into a clinical setting.
It was a pleasure to read this work.
Author Response
Dear Reviewer 2,
Thank you very much for your careful reading and kind suggestions.
The uploaded manuscript is a revised version of our research article,
Title: Quantitative Analysis of Factors Regulating Angiogenesis for Stem Cell Therapy
Authors: Takahiro Shimazaki, Nobuhiro Noro, Kazuhiro Hagikura, Taro Matsumoto, Chikako Yoshida-Noro *
Submitted to the section: Cardiovascular Biology, Stem Cells for Cardiovascular Biology and Medicine
One version is highlighted with green letters for revision. The other version is the final one in all black letters.
Comments and Suggestions for Authors:
The manuscript has been improved and spelling errors were corrected.
Further clarification of the angiogenic effects of DFAT and the precise signaling mechanism will help for researchers and clinicians to translate them into a clinical setting.
It was a pleasure to read this work.
Thank you for your agreement to accept my paper for publication.
Sincerely yours,
Chikako Yoshida-Noro, Ph.D
Division of Cell Regeneration and Transplantation, Department of Functional Morphology, Nihon University School of Medicine, 173-8610 Tokyo, Japan
Reviewer 3 Report
The authors have failed to address the comments properly.
There is no statistical analysis presented in this study. Only standard deviations are shown in the supplemental data. How do we know if the data shown is statistically significant or not?
In the revised manuscript, the authors mentioned that gene expression of huvec cells was analyzed from the indirect co-culture. However, nothing was mentioned about indirect co-culture in their methods causing more confusion and justifying my earlier comment that the methods are not clear.
The angiogenesis kit KZ-1000, KURABO, cannot be found on the KURABO website. The authors should provide more details regarding this kit where it seems like the cells (both Huvec and fibroblasts) and matrix components are preloaded in 24 well plates. This also raises additional concerns, although they eliminated the possible contamination from DFAT using indirect co-culture, the huvecs and fibroblasts are together and as a result all the RNA extracted from these cultures is combination of both huvecs and fibroblasts and not precisely huvecs and authors have no control on what percentage of huvecs and fibroblasts are present and where their RNA is primarily coming from.
The authors responded that "They were tested separately, so there were some errors, but the trends were the same." This is not true with the trends, especially if the data was normalized with the respective controls then the it would show what the authors wanted to see resulting inaccurate interpretation of the data.
If the DFAT co-culture promoted angiogenesis then it would be a best model to test the suppression properties of the polyphenols and flavonoids. Not sure why they don't want to test it.
Author Response
Dear Reviewer 3,
Thank you very much for your careful reading and kind suggestions.
The uploaded manuscript is a revised version of our research article,
Title: Quantitative Analysis of Factors Regulating Angiogenesis for Stem Cell Therapy
Authors: Takahiro Shimazaki, Nobuhiro Noro, Kazuhiro Hagikura, Taro Matsumoto, Chikako Yoshida-Noro *
Submitted to the section: Cardiovascular Biology, Stem Cells for Cardiovascular Biology and Medicine
One version is highlighted with green letters for revision. The other version is the final one in all black letters.
According to your suggestions, we revised the paper. Our responses are as follows, shown in red letters.
Comments and Suggestions for Authors:
The angiogenesis kit KZ-1000, KURABO, cannot be found on the KURABO website. The authors should provide more details regarding this kit where it seems like the cells (both Huvec and fibroblasts) and matrix components are preloaded in 24 well plates. This also raises additional concerns, although they eliminated the possible contamination from DFAT using indirect co-culture, the huvecs and fibroblasts are together and as a result all the RNA extracted from these cultures is combination of both huvecs and fibroblasts and not precisely huvecs and authors have no control on what percentage of huvecs and fibroblasts are present and where their RNA is primarily coming from.
In the following past papers using Kurabo KZ-1000, there is no description of the number of cells, and it is stated that the experiment was conducted according to the manufacturer's protocol.
https://www.ncbi.nlm.nih.gov/pmc/articles/PMC7280264/
https://www.spandidos-publications.com/or/41/6/3508?text=fulltext
https://www.ncbi.nlm.nih.gov/pmc/articles/PMC5366987/
https://acsjournals.onlinelibrary.wiley.com/doi/full/10.1002/cncr.20818
https://www.jbc.org/article/S0021-9258(19)82455-9/fulltex
In some cases, the paper by Bishop et al was cited as the original, but they used different cells, and the number of cells was not stated in this paper either.
https://pubmed.ncbi.nlm.nih.gov/14517413/
There was no experiment for gene expression analysis from the cells of 24 wells shown in those papers.
This kit is no longer commercially available.
There is no statistical analysis presented in this study. Only standard deviations are shown in the supplemental data. How do we know if the data shown is statistically significant or not?
In the above papers, a p-value was shown for comparison between the samples at one concentration. However, our experimental results showed a concentration dependency for individual samples, when a value of VEGF alone was set as 100. Statistically, we think the mean and standard deviation which is shown in the Supplemental Figures is sufficient. The other two reviewers agreed to this. We added the following sentences to the figure legend of supplement figures 2 and 3.
The results are presented as the mean±S.D. values of n=3.
In the revised manuscript, the authors mentioned that gene expression of huvec cells was analyzed from the indirect co-culture. However, nothing was mentioned about indirect co-culture in their methods causing more confusion and justifying my earlier comment that the methods are not clear.
We did not use an indirect co-culture system in this paper. We referenced Watanabe et al [12] for gene expression analysis for indirect co-culture of DFAT and MS1. But for better understanding, we changed Figure 7 using only wells with soluble factors for RT-PCR. Also, we removed the description of the gene expression analysis for co-culture in the text.
The authors responded that "They were tested separately, so there were some errors, but the trends were the same." This is not true with the trends, especially if the data was normalized with the respective controls then the it would show what the authors wanted to see resulting inaccurate interpretation of the data.
We reorganized Figure 7 to show the results of promoting and suppression experiments separately and change the photo for better understanding.
If the DFAT co-culture promoted angiogenesis then it would be a best model to test the suppression properties of the polyphenols and flavonoids. Not sure why they don't want to test it.
Experiments for promoting angiogenesis and suppressing angiogenesis were performed separately. We would like to consider the experiment you mentioned next time.
We hope the revised version will be suitable for publication and look forward to hearing from you regarding our submission.
Thank you in advance for your kind consideration of this paper.
Sincerely yours,
Chikako Yoshida-Noro, Ph.D
Division of Cell Regeneration and Transplantation, Department of Functional Morphology, Nihon University School of Medicine, 173-8610 Tokyo, Japan
Round 3
Reviewer 3 Report
The authors did not make required changes and addressed the reviewers comments.
Author Response
Dear Reviewer 3,
Thank you very much for your careful reading and kind suggestions.
Academic Editor Notes
I think the authors responded to the reviewer 3 appropriately except one comment of statistical analysis for the supplemental figure. If statistical analysis would be performed, I think this manuscript is publishable.
According to your kind instructions, we corrected the manuscript.
We performed statistical analysis and changed Figure S2 and S3, some description in Results, and Figure legends. Figures 3 to 7 and their legends were modified for the better understanding.
The uploaded manuscript is a revised version of our research article,
One version pdf is highlighted with red letters for revision 3. The other version is the final one in all black letters.
We hope the revised version will be suitable for publication and look forward to hearing from you regarding our submission.
Thank you in advance for your kind consideration of this paper.
Sincerely yours,
Chikako Yoshida-Noro, Ph.D
Division of Cell Regeneration and Transplantation, Department of Functional Morphology, Nihon University School of Medicine, 173-8610 Tokyo, Japan